# Nutrition Related Stress Factors Reduce the Transfer of Extended-Spectrum Beta-Lactamase Resistance Genes between an *Escherichia coli* Donor and a *Salmonella* Typhimurium Recipient In Vitro

**DOI:** 10.3390/biom9080324

**Published:** 2019-07-31

**Authors:** Eva-Maria Saliu, Marita Eitinger, Jürgen Zentek, Wilfried Vahjen

**Affiliations:** Freie Universität Berlin, Institute of Animal Nutrition, Königin-Luise-Str. 49, 14195 Berlin, Germany

**Keywords:** extended-spectrum β-lactamases, horizontal gene transfer, minerals, short-chain fatty acids, organic acids, feed additives, osmolarity, bacterial conjugation

## Abstract

The transfer of extended spectrum β-lactamase (ESBL)-genes occurs frequently between different bacteria species. The aim of this study was to investigate the impact of nutrition related stress factors on this transfer. Thus, an *Escherichia coli* donor and a *Salmonella* Typhimurium recipient were co-incubated for 4 h in media containing different levels of the stress factors’ pH, osmolality, copper, zinc and acetic, propionic, lactic, and n-butyric acid, as well as subtherapeutic levels of cefotaxime, sulfamethoxazole/trimethoprim, and nitrofurantoin. Conjugation frequencies were calculated as transconjugants per donor, recipient, and total bacterial count. A correction factor for the stress impact on bacterial growth was used. Acetic, lactic, and n-butyric, acid, as well as pH, showed no significant impact. In contrast, increasing concentrations of propionate, zinc, copper, and nitrofurantoin, as well as increased osmolality reduced conjugation frequencies. Sulfamethoxazole/trimethoprim and cefotaxime showed increased transconjugants per donor, which decreased after correction for stress. This study showed, for the model mating pair, that conjugation frequencies decreased under different physiological stress conditions, and, thus, the hypothesis that stress factors may enhance conjugation should be viewed with caution. Furthermore, for studies on in vitro gene transfer, it is vital to consider the impact of studied stressors on bacterial growth.

## 1. Introduction

As extended-spectrum β-lactamase (ESBL)-producing *Enterobacteriaceae* (ESBL-PE) pose a major hazard on public health, the development of methods to reduce their occurrence has gained high priority [1,2]. In livestock, the highest ESBL-PE prevalence was observed in poultry with *Escherichia coli* being the most common species [3]. Besides comprising various pathogenic isolates, nonpathogenic *E. coli* isolates may also inhabit the intestinal tract of broilers as part of the commensal microbiota [4,5], and ESBL-production is not correlated with virulence. Thus, normally no symptoms are observed in animals colonized by ESBL-producing *E. coli* (ESBL-EC). This indicates that ESBL-EC may be harmless and does not require antibiotic treatment. However, as ESBL-genes are frequently located on plasmids, which are commonly transmitted between En*terobacteriaceae* [6], these plasmids can easily be transferred to pathogenic *Enterobacteriaceae*. This can induce the spread of antibiotic resistant pathogens in livestock.

Thus, animals may develop a disease, which is difficult to treat by antibiotics. Furthermore, treatment with β-lactam antibiotics may additionally increase the ESBL-PE load by selective pressure [5,7,8]. ESBL-PE transfer from livestock to humans and companion animals may occur via direct contact, through animal products or the environment [9,10,11]. In this way, humans may obtain harmless ESBL-PE, which then could transfer their ESBL-carrying plasmids to human pathogens in the hosts’ intestine and lead to serious, difficult to treat infections. It has been shown that conjugation occurs and transconjugants persist for many generations independent of the selective pressure by antibiotics [12]. 

Bacteria, which transfer mobile genetic elements to other bacteria, are referred to as donors, while bacteria capable of acquiring the genetic information are termed recipients. Once the plasmid is successfully transferred, the recipient, harboring the ‛new’ plasmid, will be termed transconjugant [13]. As the number of transconjugants is negligible compared to donor and recipient counts, it is not differentiated between ‛conjugated’ or ‛nonconjugated’ recipients. When quantifying and comparing conjugation events, one frequently refers to conjugation frequency (CF) [14]. There are different ways to calculate CF using the donor or recipient count as reference:(1)$$\mathrm{Conjugation}\;\mathrm{frequency}\;\mathrm{based}\;\mathrm{on}\;\mathrm{donor}\;\mathrm{count}\;(\mathrm{CF}(D))\;=\;\frac{Transconjugants/mL}{Donors/mL},$$
[15,16,17].
(2)$$\mathrm{Conjugation}\;\mathrm{frequency}\;\mathrm{based}\;\mathrm{on}\;\mathrm{to}\;\mathrm{recipient}\;\mathrm{count}\;(\mathrm{CF}(R))\;=\;\frac{Transconjugants/mL}{Recipients/mL},$$
[18,19]. It must be considered for studies on stress response that these methods of calculation neglect the impact of the used stressor on the growth of the different participants. This effect may give a false impression that conjugation was influenced by supplementation while the change in conjugation frequency just mirrors a change in bacterial growth. This bias may lead to wrong assumptions [14]. To avoid this misinterpretation, a different method to calculate the conjugation efficiency (η) can be applied as
(3)$$\eta \approx \frac{T}{RD\Delta t},$$
where *T* stands for the number of transconjugants/mL, *R* for the number of recipients/mL, D for the number of donors/mL, and Δt represents the conjugation time in hours [14]. The impact of time on conjugation must not be underestimated, since conjugation frequencies may change dramatically within hours [20]. When calculating conjugation efficiency, the impact of Δt on η can be neglected due to the tremendous numeric difference between time and transconjugants, recipients, and donors. While *T*, *D*, and *R* will range between 10^2^ and 10^9^, in most experiments the time will stay below 100. Thus, mathematically, time differences will not be large enough to affect the result. Additionally, these results are not easily compared with the results calculated with the more traditional approach of dividing transconjugants by donor or recipient counts. Furthermore, a prerequisite for this equation is that donor and recipient concentrations remain rather constant during the period of time. This assumption does not apply for the intestinal microbiota. In the intestinal tract, bacteria undergo dynamic growth states with a constant change of the composition [21,22]. In this way, both pathogens and nonpathogenic bacteria, donors, and recipients share a timely and spatially dynamic habitat in animals. As they may interact, their growth and total amount will be influenced by environmental factors in different ways [5]. 

Strain dependent differences are frequently observed when conjugation trials are performed with various donor and recipient strains, as the environment and their interaction influences them differently [23]. In this sense, stress can be defined as a potential threat to the survival of the bacterial cell [24]. The mechanisms explaining the influence of stress on conjugation are manifold. Hence, stressors may influence (1) the bacterial genome, (2) number of plasmids per cell, and/or (3) efficiency of the plasmid transfer [25]. Thus, stress reactions caused by sub-lethal concentrations of antibiotics, may originate from an induction of the bacterial conjugation machinery and/or the stimulation of the excision of transferable genes from the donors’ chromosome [26,27]. It was previously hypothesized, but not proven, that antibiotics, which affect the cell wall of bacteria, increase transfer rates [28]. Additionally, stress caused by extreme pH, starvation, and/or organic solvents among others may influence the uptake and release of plasmids [25]. Different kinds of stressors, such as pH, antibiotics, or nutrient starvation may induce stress and DNA mutations (SOS response) in bacteria, which can affect conjugation rates positively. Thus, conjugation under a given stress condition for one strain exemplifies the general possibility of gene transfer under these conditions. 

As multi resistant bacteria rise as a problem and are a major health hazard, new reduction measurements are developed to reduce specific bacterial fractions of the intestinal microbiota in animals. Nutritional intervention steps have shown promising results to shift the microbiota towards a more desirable direction [29,30,31,32,33,34]. This may create stress for the suppressed bacteria, causing a change in their metabolic activity, including the transfer of genetic material [35,36,37]. Thus, modification of environmental conditions by certain feeds or feed additives can induce stress, which may threaten the survival of bacterial cells due to unfavorable conditions [36,37]. Therefore, conjugation rates may be influenced by specific feed additives such as copper and zinc, but also by bacterial metabolites (short chain fatty acids), as well as by factors defining the intestinal milieu (pH and osmolality), which may be changed by feed additives such as enzymes and pro- or prebiotics. Finally, it is known that antibiotic treatment has a tremendous impact on bacterial growth, providing a further stress factor investigated in this study [5,7]. 

From the considerations outlined above, this study was designed to investigate how different stress factors may affect the conjugation rates of an ESBL-producing *E. coli* donor strain and a *Salmonella* Typhimurium recipient strain. Special care was given to the analysis when referring to the donor, recipient, and total bacterial count, as well as addressing the impact of stress on bacterial growth.

## 2. Materials and Methods 

### 2.1. Strains and Cultivation Conditions

A nonpathogenic *E. coli* isolate (ESBL10682, isolated from the excreta of one day old broilers within the RESET program), harboring the *bla_CTX-M-1_* gene and belonging to the B1 subgroup, was chosen as the donor. The strain *Salmonella* Typhimurium L1219-R32 (isolated from pigs) was chosen as the recipient strain. This mating pair was revealed to be the best fit for the study design in a previous study obtaining conjugation kinetics for five potential *E. coli* donor strains and six potential *Enterobacteriaceae* recipients every 2 h for 22 h [38]. From these results, the mating pair was known to result in a conjugation frequency of approximately 10^5^ transconjugants/donor after 4 h of co-incubation (donor:recipient 1:1; 10^5^ cells/mL starting conditions) [38]. All cultures were obtained from cryo-stocks and cultured in Mueller Hinton 2 broth (Sigma-Aldrich, Chemie GmbH, Darmstadt, Germany). Culturing of the strains was done in Mueller Hinton 2 broth, supplemented with 8 µg/mL cefotaxime (CTX) (Alfa Aesar, Thermo Fisher GmbH, Schwerte, Germany) for *E. coli* incubation or 300 µg/mL sulfamethoxazole/trimethoprim (SXT) (Sigma-Aldrich, Chemie GmbH, Darmstadt, Germany) for the *Salmonella* Typhimurium strain. All strains were incubated aerobically at 37 °C.

### 2.2. Experimental Design

After the second preculture in antibiotic supplemented medium, the bacterial strains were washed twice in Phosphate Buffered Saline (PBS) (Sigma-Aldrich, Chemie GmbH, Darmstadt, Germany) and diluted to 5 × 10^8^ cells/mL. Fifty µL of each donor and recipient strain were added to 800 µL media supplemented with stress factors, as described below. After vigorous vortexing, the samples were incubated aerobically for four hours at 37 °C. The suspensions were then placed on ice, serially diluted, and spread on a MacConkey agar (Carl Roth GmbH + Co. KG, Karlsruhe, Germany) containing 8 µg cefotaxime/mL and 300 µg sulfamethoxazole/trimethoprim/mL to obtain transconjugants and on MacConkey agar without antibiotics to estimate the cell count of *E. coli* ESBL10682, *Salmonella* Typhimurium L1219-R32, and total bacterial count (TBC). In the set up with cefotaxime, MacConkey agar plates containing 8 µg cefotaxime/mL, or 300 µg sulfamethoxazole/trimethoprim/mL were used to obtain the cell count of *E. coli* ESBL10682 and *Salmonella* Typhimurium L1219-R32, respectively. The conjugation frequency was calculated with respect to the donor, the recipient, and the total bacterial count by dividing the number of transconjugants/mL by the respective bacteria count:(4)$$\mathrm{Conjugation}\;\mathrm{frequency}\;\mathrm{based}\;\mathrm{on}\;\mathrm{donor}\;\mathrm{count}\;(\mathrm{CF}(D))\;=\;\frac{Transconjugants/mL}{Donor\;cells/mL},$$
(5)$$\mathrm{Conjugation}\;\mathrm{frequency}\;\mathrm{based}\;\mathrm{on}\;\mathrm{recipient}\;\mathrm{count}\;(\mathrm{CF}(R))\;=\;\frac{Transconjugants/mL}{Recipient\;cells/mL},$$
(6)$$\mathrm{Conjugation}\;\mathrm{frequency}\;\mathrm{based}\;\mathrm{on}\;\mathrm{total}\;\mathrm{bacterial}\;\mathrm{count}\;(\mathrm{CF}(T))\;=\;\frac{Transconjugants/mL}{Donor+\;Recipient\;cells/mL}.$$

All experiments were repeated three times with fresh cultures and with three replicates per repetition.

### 2.3. Stress Factors

For the challenge experiments, various stress factors were added to Mueller Hinton 2 broth in different concentrations. The studied stress factors were pH, osmolality, antibiotics at subtherapeutic concentrations, zinc, copper, and the short chain fatty acids acetic, propionic, and n-butyric acid and d/l-lactate. 

#### 2.3.1. pH

The impact of pH 4–7.5 on donor and recipient growth was determined by measuring turbidity during incubation in a micro titer plate reader (Infinite200Pro, Tecan Austria GmbH, Grödig, Austria) at 690 nm every 5 min over a time period of 4 h (data not shown). A pH adjustment to 5.0, 5.5, 6.0, and 6.5 was carried out in a double concentrated Mueller Hinton 2 broth using 1 M hydrochloric acid (HCl) (Carl Roth GmbH + Co. KG, Karlsruhe, Germany). Equal volumes were achieved by adding ultrapure water to the solutions in volumetric flasks. The media were then sterile-filtered (0.2 µm, VWR International GmbH, Darmstadt, Germany). Mueller Hinton 2 broth exhibited a pH value of 7.5 and served as the control. 

#### 2.3.2. Osmolality

Sodium chloride (NaCl) (Carl Roth GmbH + Co. KG, Karlsruhe, Germany) was added to a 50 mL Mueller Hinton 2 Medium in Afnor bottles to obtain osmolalities of 200, 300, 400, 500, 600, 700, 800, 900, and 1000 mOsm/kg and autoclaved. The correct osmolality was confirmed with a micro osmometer (type OM 806, Vogel Medizinische Technik und Elektronik, Fernwald, Germany), and the impact on bacterial growth was monitored by a turbidity measurement at 690 nm. Correlation between osmolality and CF was analyzed using the software IBM SPSS (Version 22, IBM Deutschland GmbH, Ehningen, Germany). The osmolality of 300 mOsm/kg served as the control, as it resembled the osmolality of the pure medium.

#### 2.3.3. Antibiotics

Subtherapeutic levels of nitrofurantoin (F) (Sigma-Aldrich, Chemie GmbH, Darmstadt, Germany), cefotaxime, and sulfamethoxazole/trimethoprim were determined for the donor and recipient strains by studying their growth kinetics in the presence of different antibiotic concentrations. The impact on bacterial growth was monitored by turbidity measurement at 690 nm, measured for 4 h as described above. The antibiotics were added to Mueller Hinton 2 broth at each three different concentrations (0.4, 0.5, and 0.6 µg CTX/mL; 1.0, 2.5, and 5.0 µg SXT/mL; 2.0, 4.0, and 6.0 µg F/mL) while the Mueller Hinton 2 broth without antibiotics served as control. 

#### 2.3.4. Zinc and Copper

Saturated solutions of zinc from ZnO (Sigma-Aldrich, Chemie GmbH, Darmstadt, Germany) and copper from CuSO_4_(H_2_O)_5_ (Merck KGaA, Darmstadt, Germany) were prepared according to Liedtke and Vahjen [39]. Atomic absorption spectroscopy (contrAA 700, Analytic Jena AG, Jena, Germany) was used to determine actual metal concentrations. The media were then serially diluted in Mueller Hinton 2 broth and donor and recipient growth was obtained by measuring turbidity at 690 nm for 4 h. The concentrations (zinc: 0, 10, 21, 42, 84, 167 µg/mL; copper: 0, 11, 22, 43, 87, 173 µg/mL) were chosen due to their ability to reduce, but not inhibit, bacterial growth. Pure Mueller Hinton 2 broth served as the control. 

#### 2.3.5. Short Chain Fatty Acids and Lactate

Acetic acid (Carl Roth GmbH + Co. KG, Karlsruhe, Germany), propionic acid (Merck KGaA, Darmstadt, Germany), d/l-lactic acid (d/l: equal volume units, Carl Roth GmbH + Co. KG, Karlsruhe, Germany) and n-butyric acid (Sigma-Aldrich, Chemie GmbH, Darmstadt, Germany) were added to double concentrated Mueller Hinton 2 Broth. The pH was adjusted to pH 7.5 using 5 M sodium hydroxide (NaOH) (Carl Roth GmbH + Co. KG, Karlsruhe, Germany). The dilutions were sterile-filtered (0.2 µm) and the concentrations were confirmed by gas chromatography (Agilent 6890N, Agilent Technologies Deutschland GmbH, Waldbronn, Germany). Four different concentrations were prepared by 1:2 serial dilutions in Mueller Hinton 2 Broth (acetate, propionate, n-butyrate: 0, 18.75, 37.50, 75.00, 150.00 mM; lactate: 0, 13.75, 27.50, 55.00, 110.00 mM) and the exact concentrations obtained by gas chromatography. The impact on donor and recipient growth was studied prior to the conjugation experiment ). The control medium was nonsupplemented Mueller Hinton 2 Broth.

### 2.4. Calculation of Stress Impact Factor

Since stress does not only influence the conjugation but also the growth of the transconjugants, donors, and recipients, the results were corrected by a stress impact factor. This factor was defined as the percentage change of growth between the two concentrations/levels. At first, the stress impact factor (SIF), defined as the ratio between the mean colony forming units per mL of the control (cfu_ctr_) and a certain level of supplementation with a stress factor (cfu_stress_) was determined as
(7)SIF = mean cfuctrmLmean cfustressmL.

Thus, SIF = 1 would indicate no impact on the growth, SIF > 1 shows a reduction in growth, and SIF < 1 designates an enhanced growth when exposed to the stressor. Secondly, the growth of donors, recipients, and transconjugants were corrected (cfu_corr_) to the level of the controls. Thus, a condition without an impact of the stressor on bacterial growth was simulated by multiplying the cfu_stress_ with the SIF: cfu_corr_ = cfu_stress_ × SIF.(8)

In this way, cfu_corr_ was calculated for donor, recipient, total bacterial count, and transconjugants. For transconjugants, SIF was calculated according to recipient growth in all cases except when cefotaxime was supplemented, since the conjugation was assumed not to transfer growth benefits in the other cases. Conjugation frequencies were subsequently calculated as described above.

### 2.5. Statistics

All statistics were calculated with the software IBM SPSS (Version 22). Results are presented as mean values ± standard deviation. The nonparametric Kruskal–Wallis test and Mann–Whitney test were used to determine significant differences and subgroups, respectively. Differences were considered statistically significant at *p* < 0.05 and *p-*values between 0.05 and 0.1 were accepted as trends. 

## 3. Results

### 3.1. pH

A preliminary screening of the donor and recipient growth kinetics at various pH values failed to detect a significant impact on the donor or recipient strains at pH levels of 5.0–7.5. The lowest numeric conjugation frequency was noted at an initial pH of 6.0 for all alternative calculations, but differences were only marginal (Table 1). Bacterial growth was not significantly affected between pH 5.0–7.5 for all incubations (Appendix A). Thus, CF values calculated acknowledging (CF(D)(SIF), CF(R)(SIF), CF(T)(SIF)) or neglecting (CF(D), CF(R), CF(T)) the stress impact factor are rather similar (Table 1).

### 3.2. Osmolality

The growth of *Salmonella* Typhimurium declined with increasing osmolality, while the *E. coli* strain showed the highest number of colony forming units at 500 mOsm/L (Appendix A). Conjugation frequencies declined exponentially with increasing osmolality (correlation analyses: single, 3 parameters; *R*^2^ = 0.97, *R*^2^ = 0.48 and *R*^2^ = 0.96 respectively) (Table 2). After correcting for stress impact on growth, the correlation of CF(D) and CF(T) with osmolality became more linear (*R*² = 0.48, *R*² = 0.51). Significant differences in the CF of 0.4–0.5 log cfu/mL were observed for all three approaches when CF was corrected by SIF (Table 2).

### 3.3. Antibiotics

Cefotaxime had a relatively strong negative impact on the growth of the recipient strain, causing a reduction of 1.8 log cfu/mL, when no supplementation was compared to the highest CTX concentration of 0.6 µg/mL. The donor strain was less sensitive towards the substance, resulting in a reduction of 1.2 log cfu/mL from the control to 0.6 µg CTX/mL (Appendix A). When challenged with subtherapeutic concentrations of cefotaxime, the conjugation frequencies showed an increasing trend (*p* = 0.06), with increasing concentrations of CTX when calculated based on donor count. This effect was profound for CF(R) and CF(T) (Table 3). When corrected for the stress impact factor, a significant difference was observed for 0.4 µg CTX/mL supplementation in CF(D)(SIF) and CF(T)(SIF), while CF(R)(SIF) showed no significant differences.

Increasing concentrations of sulfamethoxazole/trimethoprim decreased the growth of the donor strain by 1.8 log, while the growth of the recipient strain only increased with 0.2 log from no to 5 µg SXT/mL supplementation (Appendix A). Furthermore, a strong impact on conjugation frequencies relating to donor, recipient, and total bacterial count was observed (Table 3). While the conjugation frequency increased significantly for 2.5 and 5.0 µg SXT/mL when based on the donor strain, it decreased significantly when referring to recipient and total bacterial cell growth. When corrected for the stress impact on bacterial growth, a significant decrease was observed for CF(D)(SIF), CF(R)(SIF), and CF(T)(SIF) between the control and the highest SXT concentration with 1.3, 0.9, and 1.2 log, respectively (Table 3).

Donor and recipient growth declined in a dose-dependent fashion at 0.7 and 0.8 log cfu/mL, respectively, with increasing concentrations of nitrofurantoin (Appendix A). Similarly, conjugation frequencies (CF(D), CF(T), and CF(D)) were significantly affected by the supplementation of 6.0 µg F/mL (Table 3). This effect was slightly increased when the stress impact on bacterial growth was considered. Overall, the decrease in CF ranged from 1.3 to 1.5 log.

### 3.4. Zinc and Copper

The normal growth of *E. coli* ESBL10682 was not affected significantly by zinc supplementation, while *Salmonella* Typhimurium L1219-R32 declined by 0.6 log cfu/mL at the highest investigated zinc concentration compared to the controls (Appendix A). The opposite occurred when copper was supplemented. While the recipient strain was not significantly influenced, the growth of the donor decreased with 1.5 log cfu/mL (Appendix A). Values for the conjugation frequency declined with higher concentrations of zinc and copper. Zinc concentrations of 321 and 642 μM resulted in a slight but significant decrease when CF referred to recipient counts (Table 4). When corrected for the SIF, this effect was observed for CF(D)(SIF), CF(R)(SIF), as well as CF(T)(SIF). Copper supplementation decreased CF(D), CF(R), and CF(T) significantly by 1.1, 3.2, and 2.3 log cfu/mL, respectively. A correction for the stress impact on bacterial growth showed even more severe reductions of 3.1, 3.2, and 3.2 log cfu/mL, respectively (Table 4, Figure 1). 

### 3.5. Short Chain Fatty Acids

The highest negative impact on the growth of both strains was observed in the presence of propionic acid with a decrease of 0.6 and 0.5 log units for the donor and recipient strains, respectively (Appendix A). Acetic and n-butyric acid reduced bacterial growth by approximately 0.2 and 0.4 log units, respectively (Appendix A). In the presence of lactic acid, the growth of both donor and recipient strains remained rather constant (Appendix A). Similarly, n-butyric and lactic acid had no significant impact on conjugation frequencies (Table 5). However, supplementation of media with acetic or propionic acid led to a significantly negative impact on conjugation events. Thus, acetic acid showed significantly lower CF(D)-values, but after correction for SIF, this effect disappeared (Table 5). Propionic acid supplementation, on the other hand, led to decreasing conjugation frequencies for CF(D), CF(R), and CF(T) with 0.6, 0.5, and 0.7 log (Figure 1). A correction for SIF resulted in a sharper decrease with 0.8 log cfu/mL for both CF(D)(SIF) and CF(T)(SIF), considering the difference between control and highest propionic acid concentration (Table 5). 

## 4. Discussion

The aim of this study was to investigate the impact of nutrition related stress factors on conjugation frequencies in an in vitro trial with an ESBL-producing *E. coli* donor strain and a *Salmonella* Typhimurium recipient.

Conjugation frequency (CF) is frequently calculated by dividing the number of transconjugants/mL by the donor count per mL [40,41,42,43,44,45]. However, CF can also refer to the recipient instead of donor count [12,43]. Both donor and recipient growth are generally considered independent of each other but undergo dynamic growth states in the intestine. Therefore, this study was designed to investigate the impact of both donor and recipient, as well as the total bacterial count on conjugation frequencies. The rationale behind this approach is the view that in vivo bacteria are under constant stress, and, therefore, different stressors affect the physiological response of both donor and recipient. Thus, the results of this study differ depending on the method of calculation as they address different questions. To evaluate the risk of transmission following an infection with ESBL-producing *Enterobacteriaceae*, it is important to know, how many ESBL-producing *E. coli* cells transfer their plasmid to a potential recipient. However, the health-related risks and clinical importance of antibiotic resistant pathogens may be better characterized by their uptake of resistance genes. 

The impact of different agents on conjugation events is commonly displayed as a change in CF. However, these agents do not only impact the formation of transconjugants, but also their growth and viability, as well as the growth and viability of recipient and donor. This creates a bias neglected by the common methods to calculate the CF as transconjugants/donor or transconjugants/recipient [14]. If the donor and recipient concentrations remain steady during a certain period of time, one may, therefore, refer instead to conjugation efficiency (Equation 4) [14,27]. This, however, did not apply for the current study and does not resemble the environment in the gastrointestinal tract, where changes in diet, treatment, or infection alter microbial composition [4,44,45]. Therefore, we developed a method to monitor conjugation frequencies for donor, recipient, and transconjugants under the effect of different stressors. By calculating the relative impact on growth and viability and multiplying this factor by their respective cfu/mL, all parts of the in vitro system were corrected for the growth impact of the stressor itself.

From the above it is concluded that the method of calculation has a significant impact on the results, and, thus, different methods should be considered in studies on the impact of stressors on conjugation frequency. 

In the gastrointestinal tract (GIT) of poultry, pH levels range mainly between 5 and 8, if gizzard and proventriculus are neglected [46,47]. The crop and caecum are the compartments with the highest bacterial density, where bacterial interaction is most likely to occur [4,5]. Therefore, these pH values were chosen when studying the impact of pH on CF. When investigating pH as a stress factor, no impact on conjugation was observed in this study. Similarly, the growth of both donor and recipient strain was not significantly affected. In contrast, a positive impact on CF was observed in an experiment with an *E. coli* donor and a *Salmonella* Typhimurium recipient where an Inc GpI1 plasmid was transferred at a pH value of 4.3 [25]. Similar results were observed with other different *E. coli* donors and recipients [18]. Unfortunately, conjugation frequency was only based on the recipient count, and no information on the impact of the acid stress on the growth of the strains was stated. In the same study, the impact of low temperatures was investigated, showing a significantly higher impact on the conjugation rate compared to results with different pH values. When a combination of low temperatures and a pH value of 5 was tested for two different mating pairs, the effect was very similar to the impact of the low temperature alone [18]. This suggests that the pH of 5 did not have an impact on the conjugation, corresponding to the findings in this study. The increase of transconjugants at low pH levels was also observed in a study investigating the impact of HCl at 0.032–0.128 M on conjugation [18]. However, no information on conjugation frequencies or donor and recipient growth at these levels was provided. Thus, very low pH values may have an impact on conjugation, but pH levels commonly observed in the major parts of the intestinal tract may not be low enough to have an influence. 

Osmolality in the gastrointestinal tract of broilers varies between individual bowel segments. Accordingly, osmolalities of 540 (crop), 312 (gizzard), 571 (duodenum), 650–573 (jejunum), and 514–451 (ileum) mOsm/kg were reported [48], descending the intestinal tract, while osmolarities of 390 (duodenum), 430 (jejunum), and 340 (ileum) mOsm/L were observed [47]. Despite the numeric differences between these two studies, in the small intestine, the highest values were always observed in the jejunum, followed by the duodenum and ileum. In contrast to the findings from the pH setup, osmolality reduced the conjugation frequency exponentially, when calculating CF(D) and CF(T), while a more linear reduction was observed when calculating CF(R). Similar to experiments with different pH values, the donor growth was not affected by increasing osmolality, while the recipient strain showed reduced growth/viability. Thus, it appears that conjugation is more likely to occur in the gizzard or ileum as far as osmolality is concerned.

Since the ban of antibiotic growth promoters in the European Union (EC (No) 1831/2003), alternatives have especially gained importance [49]. Zinc and copper are commonly used in animal production to increase health, feed efficiency, and body weight [50]. In this study, the impact of zinc and copper on conjugation was the most profound among all tested stressors. Conjugation frequencies were reduced with approximately 2.3 and 0.8 log levels for copper and zinc respectively after correcting for stress impact on growth. Interestingly, in the zinc set up, the highest reduction occurred for medium levels of zinc supplementations contrary to experiments with copper, which showed a decreasing frequency of conjugation with increasing copper concentrations. The decrease of CF at medium zinc concentrations, followed by an increase with increasing zinc concentrations, might indicate that zinc up to a certain concentration can reduce conjugation. However, as bacterial stress increases at higher zinc concentrations, horizontal gene transfer might also be enhanced. This suggests a correlation between concentration of stressor and CF, which must not be linear. One must acknowledge that numerically, the observed decrease of CF at 321 and 642 µM Zn(II) is rather small (<1 log cfu/mL), indicating that interpretations should be discussed critically. Further studies should investigate this observation in detail. Zinc acetate (0.2 mM zinc) reduced the transfer of an ESBL-carrying plasmid from an *Enterobacter* donor to an *E. coli* recipient under detection limit [51]. As results from this study did not show such a severe impact, it must be considered that the conjugative pair used in the present study was more prone to transfer plasmids and did so in a shorter time period. Varying conjugation frequencies were also observed for ESBL-producing *Enterobacteriaceae* in the presence of different metal surfaces with an *E. coli* and a *Klebsiella pneumoniae* donor and a *E. coli* recipient [45]. While CF declined after 2 h on stainless steel, it fell below the detection limit after 2 h incubation on the copper surface. This agrees with our results, where increasing concentrations of copper led to higher reductions of conjugation rates. Similarly, different copper supplementations of CuSO_4_ and copper nanoparticles reduced conjugation frequencies [52]. From this comprehensive perspective, the copper surface should be considered a substantial high concentration. A link between the usage of zinc and copper feed additives and the occurrence of antibiotic resistance was established repeatedly [53,54,55,56,57]. Consequently, the European Medicines Agency (EMA) and European Food Safety Authority (EFSA) recommends reducing zinc and copper in animal production [58,59]. The positive impact of copper and zinc on the reduction of CF must, therefore, be weighed against the risk of increasing prevalence of antibiotic resistant bacteria by other mechanisms and treatments. 

Bacterial metabolites may also have an impact on bacterial physiology in vivo. For instance, a reduction of conjugation frequency was previously reported in the presence of lactic acid producing bacteria, such as *Streptococcus thermophilus*, *Lactobacillus fermentum*, *Lactobacillus plantarum*, and *L*actobacillus bulgar***icus* [19,60,61,62,63]. This effect was assumed to be due to their lactate production among other factors. This hypothesis could not be confirmed here, as lactate did not lead to a reduced conjugation frequency. Interestingly, while numeric differences between the control and highest lactate concentrations were negligible, n-butyrate actually showed higher conjugation rates at its highest concentration, while acetate numerically decreased the conjugation frequency. Thus, apart from pH reduction, bacterial metabolites may also play different roles for the transfer of mobile genetic elements. This may also be the case for propionic acid, which showed a negative concentration dependent effect on conjugation. Similarly, a reduction of CF(D) was observed in an experiment with *Salmonella enterica* serovar Typhimurium donor and recipient strains derived from mice in the presence of propionate [64]. As pH as a factor can be ruled out, and the growth corrected conjugation frequency also declined drastically, it can be concluded that propionate acts differently on conjugation than acetate or n-butyrate. Finally, observed effects for propionate were only significant at concentrations that exceeded the usually observed threshold in the hindgut of poultry. However, propionate is used in quite high doses for its antibacterial and antifungal properties in animal nutrition, and, therefore, further in vivo studies should investigate the effect of propionic acid as a feed additive to counteract the transfer of ESBL-carrying plasmids in *Enterobacteriaceae*. 

The determined conjugation frequency depends on two factors—The bacterial concentration and the transconjugants’ growth. The bacterial concentration shapes the chance for donor and recipients to meet close enough to perform a plasmid transfer. On the other hand, transconjugant growth directly affects the number of detected transconjugants and thus the calculation and result of CF. Hence, it would be tempting to conclude that changes in CF can be explained solely mathematically due to variations of donor, recipient, and/or transconjugant concentrations in the presence of stressors. To investigate this further, an experiment with sublethal amounts of antibiotics was designed, to reduce the growth of (a) donor, (b) recipient, or (c) both donor and recipient. Assuming that conjugation comes at no fitness cost or gain, the transconjugants should grow similar to the recipients, as they are basically identical apart from their additional plasmid harboring resistance against CTX. If the explanation for the changes in CF were solely mathematical, the following situation would arise for the mentioned scenarios:Lower numbers of donor cells would lead to a higher ratio of transconjugants per donor cell count.Lower numbers of recipient cells would lead to lower numbers of transconjugants and a decreased ratio of transconjugants per donor cell count (except for CTX supplementation, as the transconjugants grow better than the recipients).As recipient and donor are affected equally, CF will not differ significantly from control. 

This would lead to a lower number of transconjugants per donor in the osmolality and CTX experiments. This effect should be more profound for osmolality, since both the recipient count and transconjugant growth are affected negatively. CTX, on the other hand, reduces the growth of the recipient more than transconjugant growth. Thus, one expects decreasing CF(D), CF(R), and CF(T) with increasing concentrations of CTX and higher osmolality. Similarly, SXT supplementation would result in an increase of transconjugants per donor and higher CF(D), while lower CF(R) would be expected at increasing antibiotic concentration. Nitrofurantoin inhibited the growth of donors and recipients in an equal manner. Therefore, no significant differences were predicted. The results from the experiments differed from these assumptions. Thus, the changes in conjugation frequencies cannot solely be explained by changes in bacterial growth, justifying the conclusion that some stress factors may directly influence conjugation. 

Antibiotics are frequently used in livestock as therapeutics, meta- and prophylaxis and, outside the European Union, in subtherapeutic levels to enhance performance [5,10,65]. The usage of antibiotics always comes with the risk of developing antibiotic resistant bacteria due to selective pressure, especially when used at subtherapeutic levels [5,10,66]. Simultaneously, the potential threat to survival is posing stress to the microorganisms resulting in changes of metabolism and activity [24,36,37]. This may also impact conjugation [35]. The influence of gentamycin on the transfer of an ESBL-carrying plasmid from an *E. coli* donor to *E. coli* and *Pseudomonas aeruginosa* recipients had similar prerequisites as the CTX experiment, since donor growth was not affected as much as the recipient, which declined with increasing concentrations [67]. Similar results were observed in a further study investigating the impact of three different antibiotics in a conjugation experiment with resistant donors and transconjugants but sensitive recipients [68]. *Pseudomonas aeruginosa* accepted the ESBL-carrying plasmids at a higher rate only at a state where the antibiotic concentration changed from low impact to high impact on recipient growth. The results agree with the results from the CTX experiment (CF(D)). In the case of the *E. coli* recipient, CF(D) decreased with increasing gentamycin [68] supplementation, corresponding with the results from the presented nitrofurantoin experiment. However, nitrofurantoin had a different impact on donor growth. The gentamycin effect was observed at a stage of severe impact on recipient growth, and this was not considered when CF was calculated. Thus, the low detection of transconjugants may be the reason for these results, rather than an actual change in conjugation. Similar to the nitrofurantoin experiment, it was previously described that an antibiotic substance, affecting donors and recipients equally, may lead to a reduction of conjugation frequencies [68]. The impact of antibiotics on donor, recipient, and transconjugant growth over a period of time shows how incorrect assumptions on the impact of different factors easily arise [20]. Even after recipient counts fell under the detection limit, transconjugants kept rising. Simultaneously, the ratio of transconjugants/donor and transconjugant/recipient changed with time. Both amoxicillin and ampicillin reduced transconjugant counts similar to the cefotaxime experiment with increasing antibiotic concentrations [20]. CF(D) increased in the presence of sulfamethoxazole/trimethoprim. However, when corrected for growth impact, the effect was opposite, showing a significant decrease in conjugation frequency with an increasing concentration of SXT, which was also observed for CF(R)(SIF) and CF(T)(SIF). 

## 5. Conclusions

In conclusion, a negative impact on conjugation frequency was observed for osmolality, zinc, copper, and propionic acid, as well as subtherapeutic levels of antibiotics. No effects were found for pH or the bacterial metabolites lactate, acetate, or n-butyrate. Furthermore, no stressors increased conjugation frequency, and, thus, the hypothesis that stress generally increases bacterial conjugation should be viewed with caution. The results also show that, in studies focusing on stress related effects on gene transfer, the calculation of conjugation frequency should include the impact of a stressor on donor, recipient, and transconjugant. Still, it must be considered that the observed impact on conjugation frequencies might be strain specific. Future studies should, therefore, investigate if these observations can be repeated with different donor and recipient strains. In the present study, the impact on conjugation events was investigated for one stressor at a time. However, the intestinal tract of broilers combines these and further stressors. Thus, further studies should be anticipated to examine conjugation events in complex systems.

## Figures and Tables

**Figure 1 biomolecules-09-00324-f001:**
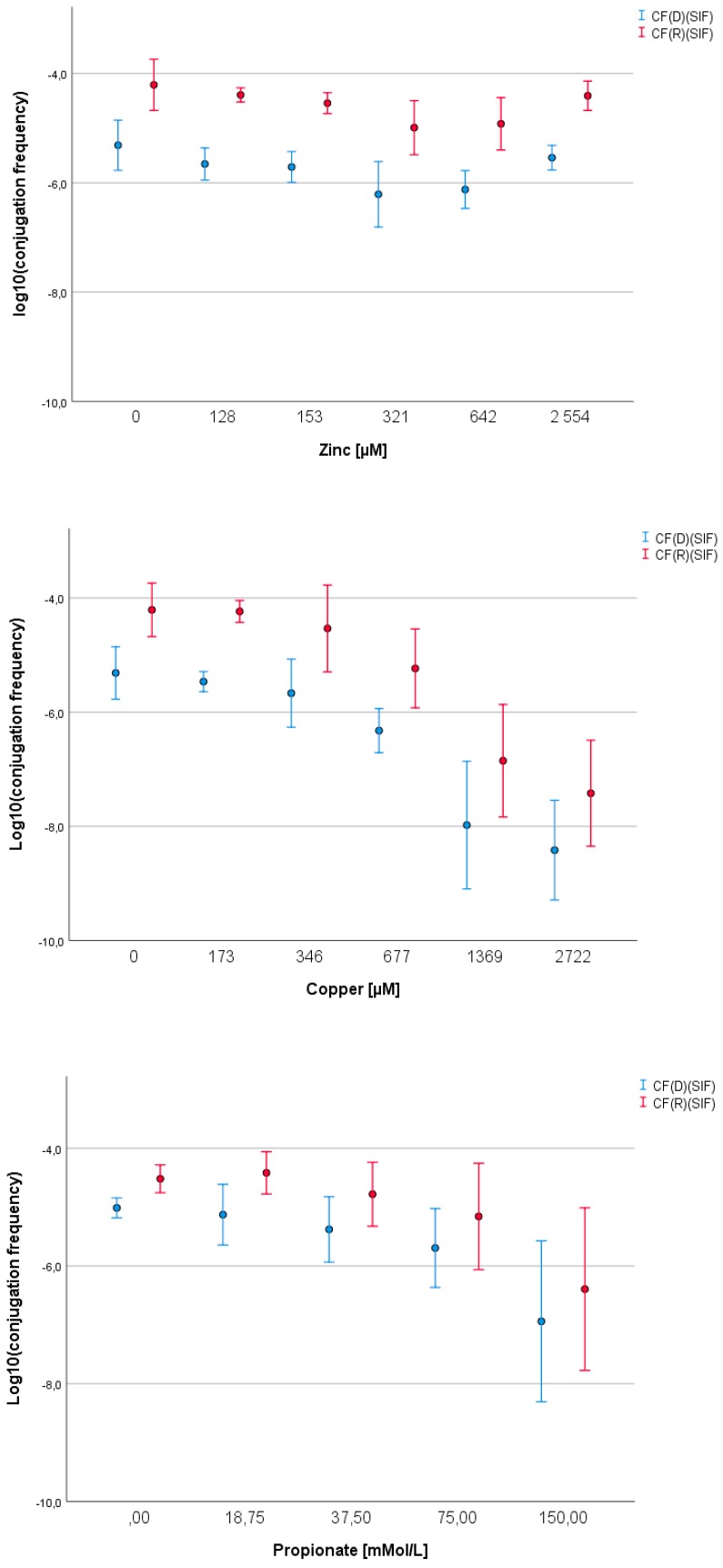
The impact of different concentrations of zinc, copper, and propionate on conjugation frequency of an *E. coli* donor and a *Salmonella* Typhimurium recipient after correction for the stress impact on bacterial growth. Mean values and standard deviations are displayed for conjugation frequencies (CF) calculated as CF(D) = transconjugants/donor or CF(R) = transconjugants/recipients; SIF = stress impact factor.

**Table 1 biomolecules-09-00324-t001:** The impact of pH on the conjugation frequency [log_10_(CF^1^)] of an *Escherichia coli* donor and a *Salmonella* Typhimurium recipient, calculated per donor, recipient, and total bacterial count corrected for the stress impact factor.

pH	CF(D)	CF(R)	CF(T)	CF(D) (SIF)	CF(R) (SIF)	CF(T) (SIF)
5.0	−5.0 ± 0.2	−4.6 ± 0.2	−5.2 ± 0.2	−5.0 ± 0.2	−4.6 ± 0.2	−5.2 ± 0.2
5.5	−5.0 ± 0.3	−4.4 ± 0.2	−5.1 ± 0.2	−5.2 ± 0.3	−4.4 ± 0.2	−5.2 ± 0.2
6.0	−5.2 ± 0.4	−4.6 ± 0.2	−5.3 ± 0.4	−5.3 ± 0.4	−4.6 ± 0.2	−5.4 ± 0.4
6.5	−5.1 ± 0.2	−4.6 ± 0.2	−5.2 ± 0.1	−5.1 ± 0.2	−4.6 ± 0.2	−5.2 ± 0.1
7.5	−5.0 ± 0.2	−4.6 ± 0.3	−5.2 ± 0.2	−5.0 ± 0.2	−4.6 ± 0.3	−5.2 ± 0.2
*p*-value	0.672	0.399	0.627	0.147	0.399	0.224

^1^ CF = conjugation frequency; CF(D) = transconjugants/donor; CF(R) = transconjugants/recipients; CF(T) = transconjugants/total bacterial count; SIF = stress impact factor (Appendix A); for each column, *p-*values were calculated comparing CF of different levels of exposure to the stressor using the nonparametric Kruskal–Wallis test and Mann–Whitney test. Raw data are provided in Appendix A.

**Table 2 biomolecules-09-00324-t002:** The impact of osmolality on the conjugation frequency [log_10_(CF^1^)] of an *E. coli* donor and a *Salmonella* Typhimurium recipient, calculated per donor, recipient, and total bacterial count corrected for the stress impact factor. The osmolality of Mueller Hinton 2 broth (control) was 300 mOsm/kg.

Osmolality (mOsm/kg).	CF(D)	CF(R)	CF(T)	CF(D) (SIF)	CF(R) (SIF)	CF(T) (SIF)
200	−5.0 ± 0.2 ^a^	−4.7 ± 0.3 ^ab^	−5.2 ± 0.2 ^ab^	−5.2 ± 0.2 ^ab^	−4.7 ± 0.3 ^ab^	−5.2 ± 0.2 ^ab^
300	−5.1 ± 0.2 ^a^	−4.6 ± 0.3 ^b^	−5.2 ± 0.2 ^a^	−5.1 ± 0.2 ^a^	−4.6 ± 0.3 ^b^	−5.2 ± 0.2 ^a^
400	−5.3 ± 0.5 ^ab^	−4.8 ± 0.6 ^bc^	−5.4 ± 0.5 ^ac^	−5.3 ± 0.5 ^a^	−4.8 ± 0.6 ^bc^	−5.4 ± 0.5 ^ac^
500	−5.3 ± 0.5 ^ab^	−4.7 ± 0.4 ^bc^	−5.4 ± 0.5 ^ac^	−5.16 ± 0.5 ^ac^	−4.70 ± 0.4 ^bc^	−5.3 ± 0.5 ^acd^
600	−5.6 ± 0.7 ^bc^	−5.3 ± 0.5 ^a^	−5.8 ± 0.7 ^b^	−5.62 ± 0.7 ^bc^	−5.25 ± 0.5 ^a^	−5.7 ± 0.7 ^bcde^
700	−5.6 ± 0.1 ^c^	−5.2 ± 0.4 ^a^	−5.8 ± 0.1 ^b^	−5.53 ± 0.2 ^bd^	−5.20 ± 0.4 ^a^	−5.7 ± 0.3 ^e^
800	−5.8 ± 0.4 ^c^	−4.9 ± 0.5 ^abc^	−5.8 ± 0.4 ^abc^	−5.52 ± 0.4 ^bd^	−4.93 ± 0.5 ^abc^	−5.7 ± 0.4 ^bce^
900	−5.7 ± 0.2 ^c^	−4.8 ± 0.1 ^abc^	−5.8 ± 0.1 ^abc^	−5.25 ± 0.2 ^abc^	−4.76 ± 0.1 ^abc^	−5.4 ± 0.2 ^abcde^
1000	−5.7 ± 0.1 ^c^	−5.0 ± 0.3 ^bc^	−5.8 ± 0.1 ^bc^	−5.57 ± 0.1 ^bd^	−4.96 ± 0.3 ^bc^	−5.7 ± 0.1 ^be^
*p-*value	<0.001	0.019	<0.001	0.005	0.019	0.003

^1^ Conjugation frequency (CF) calculated as CF/D = transconjugants/donor, CF/R = transconjugants/recipients or CF/T = transconjugants/total bacterial count; SIF = stress impact factor (Appendix A); *p-*values were calculated comparing CF of different levels of exposure to the stressor using the nonparametric Kruskal–Wallis test and Mann–Whitney test. Significant differences (*p* ≤ 0.05) between values are indicated by different superscript letters. Raw data are provided in Appendix A.

**Table 3 biomolecules-09-00324-t003:** The impact of subtherapeutic levels of antibiotics on conjugation frequency [log_10_(CF^1^)] of an *E. c*oli donor and a *Salmonella* Typhimurium recipient, calculated per donor, recipient and total bacterial count corrected for the stress impact factor.

Antibiotic ^2^ (µg/mL)	CF(D)	CF(R)	CF(T)	CF(D) (SIF)	CF(R) (SIF)	CF(T) (SIF)
**CTX**						
0	−4.9 ± 0.5	−4.3 ± 0.6 ^a^	−5.0 ± 0.5 ^a^	−4.9 ± 0.5 ^a^	−4.3 ± 0.6	−5.0 ± 0.7 ^a^
0.4	−4.4 ± 0.7	−3.0 ± 0.6 ^b^	−4.4 ± 0.7 ^b^	−5.6 ± 0.5 ^b^	−4. 9 ± 0.6	−5.7 ± 0.4 ^b^
0.5	−4.1 ± 0.7	−2.9 ± 1.0 ^b^	−4.1 ± 0.7 ^b^	−5.0 ± 0.6 ^a^	−5.0 ± 0.7	−5.3 ± 0.6 ^ab^
0.6	−4.2 ± 0.5	−2.6 ± 0.9 ^b^	−4.2 ± 0.5 ^b^	−4.7 ± 0.5 ^a^	−4.7 ± 0.6	−5.0 ± 0.5 ^a^
*p-*value	0.060	0.002	0.002	0.011	0.163	0.021
**SXT**						
0	−4.9 ± 0.5 ^ab^	−4.3 ± 0.6 ^a^	−5.0 ± 0.5 ^a^	−4.9 ± 0.5 ^a^	−4.3 ± 0.6 ^a^	−5.0 ± 0.5 ^a^
1.0	−5.6 ± 0.9 ^ac^	−5.9 ± 0.7 ^b^	−6.1 ± 0.7 ^b^	−6.9 ± 0. 9 ^b^	−5.9 ± 0.7 ^b^	−6.6 ± 0.8 ^b^
2.5	−4.3 ± 0.3 ^b^	−5.3 ± 0.4 ^b^	−5.3 ± 0.4 ^ab^	−5.7 ± 0.4 ^ab^	−5.3 ± 0.4 ^b^	−5.9 ± 0.4 ^b^
5.0	−4.1 ± 0.4 ^c^	−5.6 ± 0.5 ^b^	−5.6 ± 0.5 ^b^	−6.2 ± 0.4 ^b^	−5.6 ± 0.5 ^b^	−6.2 ± 0.5 ^b^
*p-*value	0.001	0.001	0.017	<0.001	0.001	0.001
**F**						
0	−4.9 ± 0.5 ^a^	−4.3 ± 0.6 ^a^	−5.0 ± 0.7 ^a^	−4.9 ± 0.5 ^a^	−4.3 ± 0.6 ^a^	−5.0 ± 0.5 ^a^
2.0	−5.2 ± 0.5 ^a^	−4.6 ± 0.5 ^a^	−5.3 ± 0.4 ^a^	−4.8 ± 0.7 ^a^	−4.5 ± 0.5 ^a^	−5.0 ± 0.6 ^a^
4.0	−5.2 ± 0.5 ^a^	−4.6 ± 0.3 ^a^	−5.3 ± 0.5 ^a^	−4.9 ± 0.3 ^a^	−4.5 ± 0.4 ^a^	−5.1 ± 0.3 ^a^
6.0	−6.2 ± 0.7 ^b^	−5.8 ± 0.8 ^b^	−6.4 ± 0.7 ^b^	−6.3 ± 0.6 ^b^	−5.8 ± 0.8 ^b^	−6.4 ± 0.7 ^b^
*p-*value	0.002	0.001	0.001	0.001	0.001	0.001

^1^ Conjugation frequency (CF) calculated as CF/D = transconjugants/donor, CF/R = transconjugants/recipients or CF/T = transconjugants/total bacterial count; SIF = stress impact factor (Appendix A); 2CTX = cefotaxime; SXT = sulfamethoxazole/trimethoprim; F = nitrofurantoin; *p*-values were calculated comparing CF of different levels of exposure to the stressor using the nonparametric Kruskal–Wallis test and Mann–Whitney test. Significant differences (*p* ≤ 0.05) between values are indicated by different superscript letters. Raw data are provided in Appendix A.

**Table 4 biomolecules-09-00324-t004:** The influence of zinc and copper on conjugation frequency [log_10_(CF^1^)] of an *E.* coli donor and a *Salmonella* Typhimurium recipient, calculated per donor, recipient, and total bacterial count corrected for the stress impact factor.

Minerals^2^ (µM)	CF(D)	CF(R)	CF(T)	CF(D) (SIF)	CF(R) (SIF)	CF(T) (SIF)
Zinc						
0	−5.3 ± 0.5	−4.2 ± 0.5 ^a^	−5.4 ± 0.5	−5.3 ± 0.5 ^a^	−4.2 ± 0.5 ^a^	−5.4 ± 0.5 ^a^
153	−5.2 ± 0.3	−4.5 ± 0.2 ^ab^	−5.3 ± 0.3	−5.7 ± 0.3 ^ab^	−4.5 ± 0.2 ^ab^	−5.7 ± 0.3 ^ab^
321	−5.7 ± 0.7	−5.0 ± 0.5 ^bc^	−5.8 ± 0.6	−6.1 ± 0.7 ^bc^	−5.0 ± 0.5 ^bc^	−6.1 ± 0.6 ^bcd^
642	−5.6 ± 0.4	−4.9 ± 0.5 ^bc^	−5.7 ± 0.4	−6.1 ± 0.4 ^bc^	−4.9 ± 0.5 ^bc^	−6.2 ± 0.4 ^c^
1285	−5.4 ± 0.4	−4.4 ± 0.1 ^ab^	−5.5 ± 0.3	−5.7 ± 0.3 ^abc^	−4.4 ± 0.1 ^ab^	−5.6 ± 0.3 ^abd^
2554	−5.5 ± 0.2	−4.4 ± 0.3 ^ab^	−5.6 ± 0.2	−5.5 ± 0.2 ^ab^	−4.4 ± 0.3 ^ab^	−5.6 ± 0.2 ^ab^
*p*-value	0.281	−0.015	−0.192	0.006	0.015	0.002
Copper						
0	−5.3 ± 0.5 ^a^	−4.2 ± 0.5 ^a^	−5.4 ± 0.5 ^a^	−5.3 ± 0.5 ^a^	−4.2 ± 0.5 ^a^	−5.4 ± 0.5 ^a^
173	−5.0 ± 0.2 ^a^	−4.2 ± 0.2 ^a^	−5.0 ± 0.2 ^a^	−5.5 ± 0.2 ^ab^	−4.2 ± 0.2 ^a^	−5.5 ± 0.2 ^ab^
346	−4.9 ± 0.6 ^a^	−4.5 ± 0.8 ^a^	−5.0 ± 0.6 ^a^	−5.7 ± 0.6 ^ab^	−4.5 ± 0.8 ^a^	−5.7 ± 0.6 ^ab^
677	−5.2 ± 0.4 ^a^	−5.1 ± 0.8 ^a^	−5.6 ± 0.5 ^a^	−6.3 ± 0.4 ^bc^	−5.1 ± 0.8 ^a^	−6.4 ± 0.5 ^bc^
1369	−6.4 ± 0.9 ^b^	−6.9 ± 1.0 ^b^	−7.2 ± 0.8 ^b^	−8.0 ± 1.1 ^cd^	−6.9 ± 1.0 ^b^	−8.0 ± 1.0 ^cd^
2722	−6.5 ± 0.7 ^b^	−7.4 ± 0.9 ^b^	−7.7 ± 0.7 ^b^	−8.4 ± 0.9 ^d^	−7.4 ± 0.9 ^b^	−8.6 ± 0.9 ^d^
*p*-value	<0.001	<0.001	<0.001	<0.001	<0.001	<0.001

^1^ Conjugation frequency (CF) calculated as CF/D = transconjugants/donor, CF/R = transconjugants/recipients or CF/T = transconjugants/total bacterial count; SIF = stress impact factor, ^2^ concentrations are referring to elemental zinc and copper; *p*-values were calculated comparing CF of different levels of exposure to the stressor using the nonparametric Kruskal–Wallis test and Mann–Whitney test. Significant differences (*p* ≤ 0.05) between values are indicated by different superscript letters. Raw data are provided in Appendix A.

**Table 5 biomolecules-09-00324-t005:** Influence of bacterial metabolites on conjugation frequency [log_10_(CF^1^)] of an *E.* coli donor and a *Salmonella* Typhimurium recipient, calculated per donor, recipient, and total bacterial count corrected for the stress impact factor.

Organic acid (mM)	CF(D)	CF(R)	CF(T)	CF(D) (SIF)	CF(R) (SIF)	CF(T) (SIF)
**Acetate**						
0	−5. ± 0.2 ^a^	−4.5 ± 0.2	−5.1 ± 0.2	−5.0 ± 0.2	−4.5 ± 0.2	−5.1 ± 0.2
37	−5.0 ± 0.4 ^ab^	−4.3 ± 0.5	−5.1 ± 0.4	−5.0 ± 0.4	−4.3 ± 0.5	−5.1 ± 0.4
74	−5.0 ± 0.2 ^ac^	−4.2 ± 0.6	−5.1 ± 0.2	−4.9 ± 0.1	−4.2 ± 0.6	−5.0 ± 0.1
111	−5.5 ± 0.6 ^b^	−4.6 ± 0.8	−5.6 ± 0.6	−5.4 ± 0.6	−4.6 ± 0.8	−5.5 ± 0.6
148	−5.5 ± 0.6 ^bc^	−4.8 ± 0.3	−5.6 ± 0.5	−5.3 ± 0.6	−4.8 ± 0.3	−5.4 ± 0.5
*p-*value	0.036	0.231	0.069	0.239	0.231	0.168
**Propionate**						
0	−5.0 ± 0.2 ^a^	−4.5 ± 0.3 ^a^	−5.1 ± 0.2 ^a^	−5.0 ± 0.2 ^a^	−4.5 ± 0.3 ^a^	−5.1 ± 0.2 ^ab^
36	−5.1 ± 0.5 ^a^	−4.4 ± 0.4 ^a^	−5.1 ± 0.5 ^a^	−5.1 ± 0.5 ^a^	−4.4 ± 0.4 ^a^	−5.0 ± 0.5 ^b^
73	−5.5 ± 0.6 ^ab^	−4.8 ± 0.5 ^a^	−5.6 ± 0.5 ^b^	−5.4 ± 0.6 ^a^	−4.8 ± 0.5 ^a^	−5.5 ± 0.5 ^ac^
109	−5.6 ± 0.7 ^ab^	−5.2 ± 0.9 ^ab^	−5.7 ± 0.8 ^ab^	−5.6 ± 0.7 ^ab^	−5.2 ± 0.9 ^ab^	−5.7 ± 0.8 ^ac^
145	−6.8 ± 1.4 ^b^	−6.2 ± 1.5 ^b^	−6.9 ± 1.4 ^b^	−6.9 ± 1.4 ^b^	−6.2 ± 1.5 ^b^	−7.1 ± 1.4 ^c^
*p-*value	0.002	0.006	<0.001	<0.001	0.006	<0.001
**d** **/l-Lactate**						
0	−5.0 ± 0.2	−4.6 ± 0.2	−5.1 ± 0.2	−5.0 ± 0.2	−4.6 ± 0.2	−5.1 ± 0.2
29	−5.0 ± 0.3	−4.7 ± 0.2	−5.1 ± 0.4	−5.1 ± 0.3	−4.7 ± 0.2	−5.3 ± 0.2
57	−5.2 ± 0.4	−4.7 ± 0.4	−5.4 ± 0.4	−5.2 ± 0.4	−4.7 ± 0.4	−5.3 ± 0.4
86	−5.0 ± 0.3	−4.6 ± 0.4	−5.2 ± 0.4	−5.0 ± 0.3	−4.6 ± 0.4	−5.1 ± 0.4
114	−5.0 ± 0.3	−4.6 ± 0.2	−5.2 ± 0.3	−5.1 ± 0.3	−4.6 ± 0.2	−5.2 ± 0.3
*p-*value	0.689	0.440	0.474	0.609	0.749	0.338
**n-Butyrate**						
0	−5.0 ± 0.2	−4.5 ± 0.3	−5.1 ± 0.2	−5.0 ± 0.2	−4.5 ± 0.2	−5.1 ± 0.2
38	−5.1 ± 0.4	−4.2 ± 0.7	−5.1 ± 0.4	−5.0 ± 0.4	−4.2 ± 0.4	−4.9 ± 0.4
76	−5.0 ± 0.4	−4.3 ± 0.2	−5.1 ± 0.4	−5.0 ± 0.3	−4.3 ± 0.2	−5.1 ± 0.3
114	−5.1 ± 0.5	−4.5 ± 0.2	−5.2 ± 0.4	−4.9 ± 0.5	−4.5 ± 0.2	−5.1 ± 0.4
152	−4.6 ± 0.5	−4.0 ± 0.4	−4.7 ± 0.5	−4.6 ± 0.5	−4.1 ± 0.3	−4.7 ± 0.5
*p-*value	0.439	0.116	0.442	0.302	0.102	0.320

^1^ Conjugation frequency (CF) calculated as CF/D = transconjugants/donor, CF/R = transconjugants/recipients or CF/T = transconjugants/total bacterial count; SIF = stress impact factor (Appendix A); *p-*values were calculated comparing CF of different levels of exposure to the stressor using the nonparametric Kruskal–Wallis test and Mann–Whitney test. Significant differences (*p* ≤ 0.05) between values are indicated by different superscript letters. Raw data are provided in Appendix A.

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
