# Peer review of "Nutrition Related Stress Factors Reduce the Transfer of Extended-Spectrum Beta-Lactamase Resistance Genes between an Escherichia coli Donor and a Salmonella Typhimurium Recipient In Vitro"

_biomolecules, 2019, doi:10.3390/biom9080324_

Reviewer 1 Report

1.- Table 2 which is the control  or referent conditions?

2.-  Line 328, 335,368 the authors use the word growth, I considered that is necessary  that the authors to evaluate if it correct to use growth or viability is better?

3.- Line 451, 452 and 453 the authors discuss the effect of gentamycin, however, in Material and Methods and the Results the use of  gentamycin is not indicated. Please revise

Author Response

Dear Reviewer,

thank you very much for your comments, they are most appreciated. We revised the manuscript according to your suggestions as follows:

We added the sentence ‘The osmolality of Mueller Hinton 2 broth (control) was 300 mOsm/kg.’ in the title (line 239)

Thank you for this advice. As the cultures were started with low cell numbers and increased significantly after 4 hours, we would like to refer to it as growth. Still, as you correctly point out, viability is affected as well. Thus, we changed growth to growth and viability in lines 328 and 335 and to growth/viability in line 370.

We apologise for the confusion. The effect of gentamycin was observed in a different study. To clarify this, we changed the sentence (line 451-453) to ‘In the case of the E. coli recipient, CF/D decreased with increasing gentamycin [68] supplementation, corresponding with the results from the presented nitrofurantoin experiment.’

Reviewer 2 Report

See Attached Critique.  

Author Response

Dear Reviewer,

thank you very much for your comments, they are most appreciated. We addressed them as follows:

Thank you for your comment. A manuscript (currently under review) was submitted to a peer review journal, describing the experiment, that led to the choice of mating pair. Here, conjugation kinetics were obtained for five potential E. coli donor strains and six potential donors every 2 hours for 22 hours. The mating pair presented in this study was the best fit for the chosen study design. To address this, we added ‘This mating pair was revealed best fitted for the study design in a previous study obtaining conjugation kinetics for five potential E. coli donor strains and six potential Enterobacteriaceae recipients every 2 hours for 22 hours (manuscript submitted). From these results, the mating pair was…’ (lines 109-111)

Thank you for your comment. It may seem that the donor decreased significantly due to the chosen scale of the figure. When the results are shown logarithmically, it shows that the difference is approximately 1 log.
The donor strain grew well at concentrations of up to 8 µg/mL, which is quite high regarding that the CLSI advices 1 µg/mL for the detection of ESBL-producers. Regarding table 3, only conjugation frequencies, not donor concentrations are shown.

Thank you for your advice. We added ‘2concentrations are referring to elemental zinc and copper’ to the footer of table 4

Page 7, line 247: thank you, we added ‘log’

Page 7, line 261: the sentence was adjusted as proposed

Page 8, line 300: the sentence was adjusted as proposed

Page 11, line 366: Thank you for your comment. This statement is not meant to describe the quantity, but the quality of the reduction. As we investigated nine different osmolalities, a mathematical comparison could be made for this parameter. For copper, with only 6 different concentrations, this was not calculated.

Page 12: Thank you for the comment. We revised the section as follows:

‘The determined conjugation frequency depends on to factors – the bacterial concentration and the transconjugants’ growth. The bacterial concentration shapes the chance for donor and recipients to meet close enough to perform plasmid transfer. On the other hand, the transconjugants growth directly affects the number of detected transconjugants and thus the calculation and result of CF. Hence, it would be tempting to conclude that changes in CF can be explained solely mathematically due to variations of donor, recipient and/or transconjugant concentrations in the presence of stressors. To investigate this further, an experiment with sublethal amounts of antibiotics was designed, to reduce the growth of a) donor, b) recipient or c) both donor and recipient. Assuming that conjugation comes at no fitness cost or gain, the transconjugants should grow similar to the recipients, as they are basically identical besides the additional plasmid harboring resistance against CTX. If the explanation for changes in CF would be solely mathematical, the following situation would arise for the mentioned scenarios:

lower numbers of donor cells would lead to a higher ratio of transconjugants per donor cell count

lower numbers of recipient cells would lead to lower numbers of transconjugants and a decreased ratio of transconjugants per donor cell count (except for CTX supplementation, as the transconjugants grow better than the recipients)

as recipient and donor are affected equally, CF will not differ significantly from control 

This would lead to a lower number of transconjugants per donor in the osmolality- and CTX experiments. The effect should be more profound for osmolality, since both recipient count and transconjugant growth are affected negatively. CTX on the other hand reduces the growth of the recipient more than transconjugant growth. Thus, one expects decreasing CF/D, CF/R and CF/T with increasing concentrations of CTX and higher osmolality. Similarly, SXT supplementation would result in an increase of transconjugants per donor and higher CF/D while lower CF/R would be expected with increasing antibiotic concentration. Nitrofurantoin inhibited the growth of donor and recipient in an equal manner. Therefore, no significant differences would be predicted. The results from the experiments differed from these assumptions. Thus, the changes in conjugation frequencies cannot solely be explained by changes in bacterial growth, justifying the conclusion that some stress factors may directly influence conjugation.’

Thank you very much for your comments and congratulations!